# Demarcation Line Depth in Epithelium-Off Corneal Cross-Linking Performed at the Slit Lamp

**DOI:** 10.3390/jcm11195873

**Published:** 2022-10-04

**Authors:** Farhad Hafezi, Nan-Ji Lu, Jad F. Assaf, Nikki L. Hafezi, Carina Koppen, Riccardo Vinciguerra, Paolo Vinciguerra, Mark Hillen, Shady T. Awwad

**Affiliations:** 1Laboratory for Ocular Cell Biology, Center for Applied Biotechnology and Molecular Medicine, University of Zurich, 8006 Zurich, Switzerland; 2ELZA Institute, 8953 Dietikon, Switzerland; 3USC Roski Eye Institute, University of Southern California, Los Angeles, CA 90007, USA; 4Department of Ophthalmology, Faculty of Medicine, University of Geneva, 1205 Geneva, Switzerland; 5School of Ophthalmology and Optometry, Wenzhou Medical University, Wenzhou 325015, China; 6Department of Ophthalmology, University of Antwerp, 2610 Wilrijk, Belgium; 7Department of Ophthalmology, American University of Beirut Medical Center, Beirut 1107, Lebanon; 8Department of Ophthalmology, Humanitas San Pio X Hospital, 20100 Milan, Italy; 9The School of Engineering, University of Liverpool, Liverpool L69 3BX, UK; 10Department of Biomedical Sciences, Humanitas University, Via Rita Levi Montalcini 4, 20090 Milan, Italy; 11Humanitas Clinical and Research Center–IRCCS, Via Manzoni 56, 20089 Milan, Italy

**Keywords:** corneal cross-linking, CXL, ectasia, keratoconus, cornea, slit lamp

## Abstract

We aimed to evaluate the depth of the demarcation line following accelerated epithelium-off corneal cross-linking (A-CXL) performed at the slit lamp with the patient sitting in an upright position. Twenty-three eyes from twenty patients, undergoing epi-off A-CXL (9 mW/cm^2^ for 10 min) using a CXL device at the slit lamp in the upright position. Demarcation line depth was assessed at 1 month after the procedure using anterior segment optical coherence tomography (AS-OCT) and specialized software. Surgery was uneventful in all cases. The average postoperative demarcation line depth achieved was 189.4 µm (standard deviation: 58.67 µm). The demarcation line depth achieved with patients sitting upright, receiving CXL at the slit lamp, is similar to published data on CXL performed in the supine position, suggesting that demarcation line depth is not dependent on patient orientation during CXL.

## 1. Introduction

Corneal cross-linking (CXL) involves soaking the corneal stroma with riboflavin, which is then irradiated with ultraviolet-A (UV-A) light [1]. This photoactivates the riboflavin and generates multiple reactive oxygen species (ROS) that react with stromal molecules (principally collagen) and covalently bind them (cross-link) together [2]. The principal indication is to arrest the progression of corneal ectasias like keratoconus.

Approximately two weeks after a CXL procedure, optical coherence imaging or slit lamp examination using a fine slit and maximum brightness can be used to detect changes in reflectivity in the corneal stroma, first described by Seiler and Hafezi as the “demarcation line” [3]. Whether or not the demarcation line represents a marker of the depth of effect of the CXL procedure remains to be clarified [4,5].

A recent advance in CXL is to perform the procedure at the slit lamp [6]. Historically, patients received UV-A irradiation while lying down. This study does not aim to assess the efficacy of slit lamp-based CXL, but solely to determine whether the depth of the demarcation line in slit lamp CXL is different from published values of the demarcation line in patients who received CXL in the lying position.

## 2. Materials and Methods

### 2.1. Patients

This was a retrospective analysis of individuals who underwent epi-off CXL for the treatment of progressive keratoconus, performed by a single surgeon (FH) in an office-based setting at the ELZA Institute in Zurich, Switzerland. Twenty-three eyes from 20 patients (17 males, 3 females; mean age 30.2 years, range 17–50 years) were included. Preoperative corneal thicknesses ranged from 331 µm to 556 µm, with average and median thicknesses of 442 µm and 435 µm, respectively. Twenty-two eyes had progressive keratoconus and 1 eye had ectasia after laser-assisted in situ keratomileusis (LASIK). No other ocular comorbidities were noted.

### 2.2. CXL Procedure

The CXL at the slit lamp procedure was performed as described previously by Hafezi et al. [6]. Briefly, patients received topical anesthesia with oxybuprocaine and tetracaine and were then placed at the slit lamp for seat, chinrest, and height adjustments. The periorbital region was disinfected using octenidine dihydrochloride (Octenisept, Schülke & Mayr GmbH, Norderstedt, Germany) before an open-wire Kratz speculum (C-Eye Procedure Kit; EMAGine AG, Zug, Switzerland) was placed. The epithelium was removed with a cotton swab soaked in 40% ethanol solution, before being rinsed with balanced salt solution [6]. Hypo-osmolaric 0.1% riboflavin (RIBO-KER; EMAGine AG, Zug, Switzerland) was instilled with the patient lying supine in a reclining chair, every 2 min for a total of 10 min. Excess riboflavin was rinsed off with balanced salt solution and ultrasonic corneal pachymetry was performed to determine minimal stromal thickness (SP-1000, Tomey Corporation, Aichi, Japan) with the points measured being selected with reference to pre-operative corneal topography maps. This was performed at three time points: before, after 5 min of UV-A irradiation, and immediately after irradiation was complete. The CXL device (C-eye, EMAGine AG, Zug, Switzerland) irradiated the cornea at 9 mW/cm² for 10 min (Table 1). After irradiation, topical antibiotics, and steroid drops (dexamethasone 0.1%/ tobramycin 0.3%, Tobradex, Novartis Pharma, Rotkreuz, Switzerland) and non-steroidal anti-inflammatory drops (Ketorolac 0.05%, Acular, HCI Solutions, Switzerland) were administered and a contact lens applied. Finally, the speculum was removed with care.

### 2.3. Analysis of Demarcation Line Depth

Anterior segment optical coherence tomography (AS-OCT) (MS-39, CSO Italia, Scandicci, Italy) was performed at 1 month postoperatively to evaluate the corneal stroma for the presence of the demarcation. A patented, machine learning-derived anterior segment OCT image analysis software was used to detect the demarcation line objectively and automatically (OCT Analysis, American University of Beirut, Lebanon, OCTAnalysis.com, US Patent 10,748,287). The science behind the algorithm and the clinical accuracy have been described previously [7,8,9]. The distance in pixels from the upper corneal boundary to the demarcation line was measured by the software and converted to micrometers (Figure 1). Multiple wide-field OCT sections were taken along different meridians for each eye. These scans were fed into the OCT Analysis software for automated demarcation line detection and depth computation. To improve precision, depth values were then averaged for each eye.

## 3. Results

The postoperative demarcation lines achieved by cross-linking corneas using hypo-osmolaric 0.1% riboflavin for a 10 min UV-A irradiation period using 9 mW/cm² at the slit lamp with a total fluence of 5.4 J/cm² are illustrated in Figure 2. The average postoperative demarcation line depth was 189.4 µm (standard deviation: 58.67 µm).

## 4. Discussion

Epi-off CXL using accelerated settings with 9 mW/cm² for 10 min is one of the best characterized and efficient CXL protocols besides the original Dresden protocol [14]. Here, we report that CXL performed at the slit lamp generates mean demarcation line depths in the same order of magnitude as those reported previously for CXL performed with the patient lying supine (Figure 2) [9,10,11,12,13].

A successful CXL procedure requires the following prerequisites: the presence of oxygen in the corneal stroma, irradiation with UV-A light, and the presence of riboflavin in the corneal stroma in a sufficient concentration. The diffusion of oxygen molecules into the cornea should not be affected by the relative position of the patient to the light source, whether lying or sitting upright and similarly, the propagation of photons should be independent of the patient’s orientation. Concerning riboflavin concentration in the upright position, we have published previously that gravity shows the first measurable effect on riboflavin distribution in the sitting position after 60 min, whereas irradiation takes 10 min only [15]. Consequently, demarcation line depth should therefore be unaffected by the position in which the patient is cross-linked.

One factor that does affect demarcation line depth is the vehicle used in riboflavin solutions. In general, hydroxypropyl methylcellulose (HPMC)-based riboflavin solutions result in deeper riboflavin penetration and demarcation line depth as compared to dextran solutions [16]. It is worth noting, however, that HPMC-based riboflavin solutions display greater variability in demarcation line depth when used with a 9 mW/cm²/10 min irradiation protocol, such as that reported by Pircher et al. (200 ± 99.76 µm) [10], when compared with dextran-based riboflavin solutions that display considerably lower variability in demarcation line depth (e.g., 203 ± 45 µm and 209 ± 50 µm found by Ng and colleagues [9]; 288 ± 42 µm by Kymionis and colleagues [12], and 265 ± 40 µm by Thorsrud et al. [16].

The rationale for performing CXL with the patient sitting upright at the slit lamp, rather than supine on a surgical bed, is to make the procedure easier to perform outside of an operating theater. Operating theater use carries significant staffing, administrative, and maintenance costs, and often, several surgeons compete for time slots. However, as CXL significantly reduces the microbial loads on the cornea to such an extent that it is used as an infectious keratitis treatment method, the cornea is rendered essentially sterile at the end of the procedure. CXL does not need to be performed in an operating room; it can be performed instead in a minor procedure room or even the doctor’s office, where the slit lamp is ubiquitous. In low to middle-income countries, operating theaters exist in hospitals, which are predominantly only in large cities, whereas the great majority of the population live in distant rural areas, approaches like performing CXL at the (near ubiquitous in eyecare settings) slit lamp help to not only reduce costs but also “democratizes” the procedure to a far wider population that would otherwise be very unlikely to receive this procedure.

## 5. Conclusions

This study was focused on reporting the demarcation line depth of patients undergoing an accelerated epi-off CXL protocol performed at the slit lamp. Our results suggest that the demarcation line depth is independent of the patient’s position. A comparative study of demarcation line depths in UV irradiation at the slit lamp and in the supine position using the same riboflavin solution with the same vehicle would help further evaluate these findings.

## Figures and Tables

**Figure 1 jcm-11-05873-f001:**
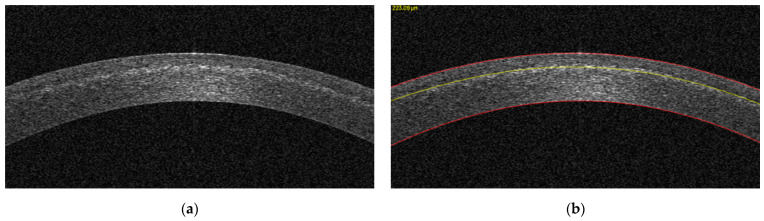
OCT image, cropped by the OCT Analysis software, at the center of the cornea (**a**), and the same OCT image after image processing (**b**). The red lines constitute the upper and lower corneal boundaries, while the yellow line represents the detected demarcation line with its depth in micrometers displayed on the top left of the image.

**Figure 2 jcm-11-05873-f002:**
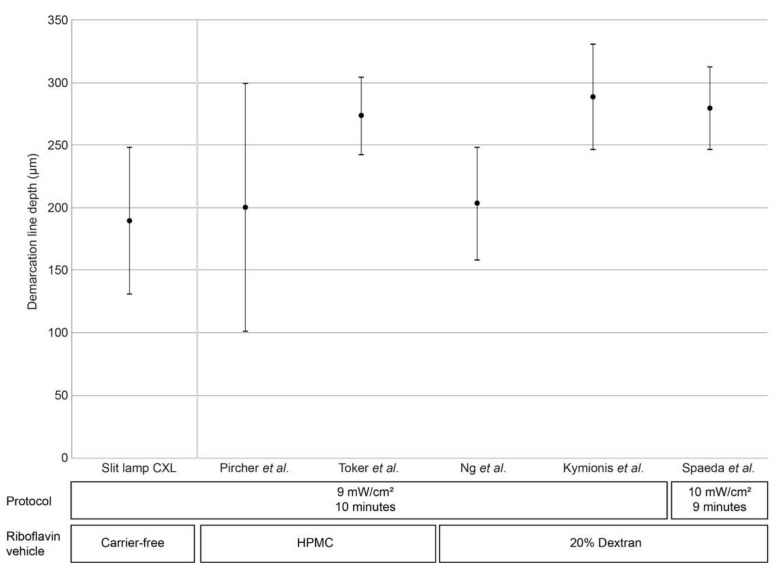
Demarcation line depth at after accelerated CXL (9 mW/cm² for 10 min) performed at the slit lamp and from previous publications that report CXL procedures using the similar irradiation settings (either 9 mW/cm² for 10 min [9,10,11,12] or 9 mW/cm² for 10 min) [13] with patients in a lying position. Black circles denote mean demarcation line depth, bars denote standard deviation; triangles denote maximum/minimum values (where available). HPMC, hydroxypropyl methylcellulose.

**Table 1 jcm-11-05873-t001:** Corneal cross-linking procedure: cross-linking device technical settings, irradiation protocols, and riboflavin used.

Parameter	Experimental Groups
Treatment target	Ectasia treatment
Fluence (total; J/cm²)	5.4
Soak Time (min)	10
Intensity (mW/cm²)	9
Treatment time (min)	10
Epithelium status	Off
Chromophore	0.1% riboflavin (RiboKER, EMAGine AG)
Light source	C-eye
Irradiation mode	Continuous

## Data Availability

Data are available on reasonable request.

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
