# Peer review of "Demarcation Line Depth in Epithelium-Off Corneal Cross-Linking Performed at the Slit Lamp"

_jcm, 2022, doi:10.3390/jcm11195873_

Round 1
Reviewer 1 Report
In this work, the authors compare the accelerated CXL demarcation depth line to previous work. The manuscript is succinct and clear, but there are a few minor adjustments that would improve its strength.
The data presentation in Figure 2 could be much stronger if the data was normalized to the corneal thickness. For example, 33% depth if the demarcation line was at 200 micrometers from a total thickness of 600 micrometers.
The patient information and demographics should also be included.
Was there a dependence on the demarcation line depth as a function of age, sex, corneal thickness, etc.?
Author Response
We would like to thank the reviewer for their thoughtful comment and input into this manuscript.
Please let us address these comments in turn.
In this work, the authors compare the accelerated CXL demarcation depth line to previous work. The manuscript is succinct and clear, but there are a few minor adjustments that would improve its strength.
We wish to thank the reviewer for this.
The data presentation in Figure 2 could be much stronger if the data was normalized to the corneal thickness. For example, 33% depth if the demarcation line was at 200 micrometers from a total thickness of 600 micrometers.
We completely understand the rationale for this comment, but in this context, we respectfully disagree. CXL is designed to strengthen the anterior cornea, and so the depth of effect is important to note, in this context, more than the percentage depth of effect. As we have shown through our work to develop an algorithm to predict the depth of cross-linking effect for the sub400 thin cornea CXL protocol, if all things are constant other than the corneal thickness during CXL (e.g. UV intensity, duration, riboflavin solution, oxygen levels, etc.) are constant, then the demarcation line/ depth of cross-linking effect should essentially be the same, irrespective of the thickness of the cornea. This means that the starting thickness of the cornea would be more important to the %depth effect than the CXL protocol used - and (in addressing a later comment) we have noted that the preoperative corneal thicknesses in this study ranged from 331-556 µm. Further, it is not clear that we could perform this analysis on the work of the other authors shown in this figure.
The patient information and demographics should also be included.
We would like to thank the reviewer for this. As alluded to in the answer above, we have added the corneal thickness range, median and average thicknesses in lines 55-58, to accompany the age range, gender, and presenting indication (progressive keratoconus)
Was there a dependence on the demarcation line depth as a function of age, sex, corneal thickness, etc.?
We agree with the reviewer that these statistics would be excellent to know, but the scope of this study was narrow, and we only had 23 eyes from 20 patients, and this is unfortunately too small to draw firm conclusions on these topics. However, we may be able to report on this across some medium at some time in the future when higher patient numbers are available for analysis.
Reviewer 2 Report
Thank you for consulting me to review this interesting manuscript. In this paper, the authors conduct a retrospective study to determine the demarcation line depth following CXL done in an upright position. I have the following comments for the authors:
1) From reviewing the manuscript, it seems as though the technique is similar to patients being supine – that is, when riboflavin was being instilled, patients were reclined supine in their chair. Is there a reason why this technique is chosen over the conventional approach of doing the entire procedure supine?
2) Although the authors reference another study for their technique, given that ‘upright CXL’ is exceedingly uncommon, it would be helpful for readers for the authors to elaborate further, especially with regards to whether this technique is epithelium-off or transepithelial.
3) Demarcation line depth has been studied in the context of accelerated versus conventional CXL, with accelerated CXL producing a shallower demarcation line. Do the authors anticipate that their upright protocol would also produce a deeper demarcation line if done without an accelerated approach?
4) Given that this study is one of the very few to evaluate this approach to CXL, it would be beneficial for readers if the authors could elaborate further on its pros and cons in their Discussion.
Author Response
We would like to thank the reviewer for their thoughtful comment and input into this manuscript.
Please let us address these comments in turn.
Thank you for consulting me to review this interesting manuscript. In this paper, the authors conduct a retrospective study to determine the demarcation line depth following CXL done in an upright position. I have the following comments for the authors:
1) From reviewing the manuscript, it seems as though the technique is similar to patients being supine – that is, when riboflavin was being instilled, patients were reclined supine in their chair. Is there a reason why this technique is chosen over the conventional approach of doing the entire procedure supine?
Indeed, we would like to thank the reviewer for this comment. The rationale of this study was to determine whether slit lamp (sitting) and supine (conventional) CXL are similar. In response to the reviewer's point #4, we have expanded in the discussion as to why we would want to perform slit lamp (sitting) CXL.
2) Although the authors reference another study for their technique, given that ‘upright CXL’ is exceedingly uncommon, it would be helpful for readers for the authors to elaborate further, especially with regards to whether this technique is epithelium-off or transepithelial.
We would like to thank the reviewer for this comment. We added the following text: "The epithelium was removed with a cotton swab soaked in 40% ethanol solution, before being rinsed with balanced salt solution [6]." to lines 67-69, in addition to the expanded discussion requested in point #4.
3) Demarcation line depth has been studied in the context of accelerated versus conventional CXL, with accelerated CXL producing a shallower demarcation line. Do the authors anticipate that their upright protocol would also produce a deeper demarcation line if done without an accelerated approach?
Thank you for this insightful comment. We would also expect this, given the points made in lines 119-128 that the physics of oxygen diffusion / UV-riboflavin-tissue interaction, and that riboflavin diffusion due to gravity is negligable for at least an hour after riboflavin saturation of the cornea.
4) Given that this study is one of the very few to evaluate this approach to CXL, it would be beneficial for readers if the authors could elaborate further on its pros and cons in their Discussion.
We have added the following text to lines 139-151:
The rationale for performing CXL with the patient sitting upright at the slit lamp, rather than supine on a surgical bed, is to make the procedure easier to perform outside of an operating theater. Operating theater use carries significant staffing, administrative, and maintenance costs, and often, several surgeons compete for time slots. But as CXL significantly reduces the microbial loads on the cornea to such an extent that it is used as an infectious keratitis treatment method, the cornea is rendered essentially sterile at the end of the procedure. CXL does not need to be performed in an operating room; it can be performed instead in a minor procedure room or even the doctor’s office, where the slit lamp is ubiquitous. In low to middle-income countries, operating theaters exist in hospitals, which are predominantly only in large cities, whereas the great majority of the population live in distant rural areas, approaches like performing CXL at the (near ubiquitous in eyecare settings) slit lamp help to not only reduce costs but also “democratizes” the procedure to a far wider population that would otherwise be very unlikely to receive this procedure.